# A Polymorph of Dipeptide Halide Glycyl-L-Alanine Hydroiodide Monohydrate: Crystal Structure, Optical Second Harmonic Generation, Piezoelectricity and Pyroelectricity

**DOI:** 10.3390/ma16103690

**Published:** 2023-05-12

**Authors:** Rosa M. F. Baptista, Clara S. B. Gomes, Bruna Silva, João Oliveira, Bernardo Almeida, Cidália Castro, Pedro V. Rodrigues, Ana Machado, Ruben B. Freitas, Manuel J. L. F. Rodrigues, Etelvina de Matos Gomes, Michael Belsley

**Affiliations:** 1Centre of Physics of Minho and Porto Universities (CF-UM-UP), Laboratory for Materials and Emergent Technologies (LAPMET), University of Minho, Campus de Gualtar, 4710-057 Braga, Portugal; rosa_batista@fisica.uminho.pt (R.M.F.B.); brunasilva@fisica.uminho.pt (B.S.); b8171@fisica.uminho.pt (J.O.); bernardo@fisica.uminho.pt (B.A.); mrodrigues@fisica.uminho.pt (M.J.L.F.R.); emg@fisica.uminho.pt (E.d.M.G.); 2LAQV-REQUIMTE, Department of Chemistry, NOVA School of Science and Technology, NOVA University Lisbon, 2829-516 Caparica, Portugal; clara.gomes@fct.unl.pt; 3UCIBIO, Department of Chemistry, NOVA School of Science and Technology, NOVA University Lisbon, 2829-516 Caparica, Portugal; 4i4HB, NOVA School of Science and Technology, NOVA University Lisbon, 2829-516 Caparica, Portugal; 5Institute for Polymers and Composites, University of Minho, Campus de Azurém, 4800-058 Guimarães, Portugal; cidaliacastro@dep.uminho.pt (C.C.); pedro.rodrigues@dep.uminho.pt (P.V.R.); avm@dep.uminho.pt (A.M.); 6Department of Electronic Engineering, Universidade do Minho, Campus de Gualtar, 4710-057 Braga, Portugal; ruben.freitas@dei.uminho.pt

**Keywords:** glycyl-L-alanine iodide, polymorphism, crystal structure, cyclic dipeptides, optical second harmonic generation, pyroelectricity

## Abstract

A polymorph of glycyl-L-alanine HI.H_2_O is synthesized from chiral cyclo-glycyl-L-alanine dipeptide. The dipeptide is known to show molecular flexibility in different environments, which leads to polymorphism. The crystal structure of the glycyl-L-alanine HI.H_2_O polymorph is determined at room temperature and indicates that the space group is polar (P2_1_), with two molecules per unit cell and unit cell parameters a = 7.747 Å, b = 6.435 Å, c = 10.941 Å, α = 90°, β = 107.53(3)°, γ = 90° and V = 520.1(7) Å^3^. Crystallization in the polar point group 2, with one polar axis parallel to the b axis, allows pyroelectricity and optical second harmonic generation. Thermal melting of the glycyl-L-alanine HI.H_2_O polymorph starts at 533 K, close to the melting temperature reported for cyclo-glycyl-L-alanine (531 K) and 32 K lower than that reported for linear glycyl-L-alanine dipeptide (563 K), suggesting that although the dipeptide, when crystallized in the polymorphic form, is not anymore in its cyclic form, it keeps a memory of its initial closed chain and therefore shows a thermal memory effect. Here, we report a pyroelectric coefficient as high as 45 µC/m^2^K occurring at 345 K, one order of magnitude smaller than that of semi-organic ferroelectric triglycine sulphate (TGS) crystal. Moreover, the glycyl-L-alanine HI.H_2_O polymorph displays a nonlinear optical effective coefficient of 0.14 pm/V, around 14 times smaller than the value from a phase-matched inorganic barium borate (BBO) single crystal. The new polymorph displays an effective piezoelectric coefficient equal to $d_{eff}=280\;pCN^{-1}$, when embedded into electrospun polymer fibers, indicating its suitability as an active system for energy harvesting.

## 1. Introduction

Glycyl-L-alanine hydroiodide monohydrate (H_2_N-CH_2_-CO-NH-CH(CH_3_)-COOH·HI·H_2_O is a dipeptide hydrohalide whose crystal structure was determined in 1989 and that is used as a model for the study of inter- and intramolecular hydrogen bonds in peptides. It is a non-centrosymmetric polar compound that crystallizes in space group P2_1_, with two molecules in the unit cell [1]. The crystal possesses a polar 2-fold axis parallel to the crystallographic b axis, and, consequently, piezoelectricity, pyroelectricity and optical second harmonic generation are allowed by symmetry. Among these properties, pyroelectricity was reported as a function of temperature in the range 100–300 K, indicating a pyroelectric coefficient along the polar axis that varies between 2 μC·m^−2^K^−1^ at 100 K and 15.5 μC m^−2^K^−1^ at 357 K [2]. The compound was synthesized at room temperature from benzyoxycarbonyl-glycyl-L-alanine methyl ester with an excess of iodine hydride (HI) [3,4]. Crystals obtained through this synthetic route are hereafter identified as Gly-L-Ala.HI.H_2_O (Poly1).

Here, we report a polymorph of glycyl-L-alanine hydroiodide monohydrate (hereafter referred to as Gly-L-Ala.HI.H_2_O (Poly2)) obtained from the synthesis of the cyclic dipeptide cyclo-glycyl-L-alanine with hydroiodic acid (HI) in an aqueous solution at room temperature. The newly discovered polymorph has unit cell parameters very similar to Poly1, but the atomic positions of the iodine and dipeptide within the unit cell are different. Although we have started from the cyclic dipeptide, its conformation in the unit cell of the Gly-L-Ala.HI.H_2_O (Poly2) is not cyclic as the initially closed dipeptide chain was broken during the synthesis process and the dipeptide has crystallized with a linear chain displaying a different conformation from that reported for Poly1.

In general, polymorphs differ either in the unit cell parameters, their space group or both. The type of polymorphism now reported for Gly-L-Ala.HI.H_2_O is very unusual, as both the unit cell parameters and the space group are the same, that is, Poly1 and Poly2, both crystalize in space group $P2_{1}$. The difference between the two polymorphs results from the dipeptide and water molecules’ positions, as well as iodine atomic positions within the unit cell displaying directionally different hydrogen bonds as a result of different atomic coordinates.

Cyclo-glycyl-l-alanine, one of the simplest dipeptides, is a chiral cyclic di-amino peptide found in nature and synthesized by some microorganisms [5,6]. The dipeptide, resulting from the bonding of glycine and l-alanine amino acids, can adapt its conformation according to a certain functional structure. This dipeptide behavior was demonstrated when used as a linker in metal–organic framework compounds [7,8,9]. We reason that the polymorph now discovered (Gly-L-Ala.HI.H_2_O (Poly2)) starting from the cyclic dipeptide in an acidic aqueous solution, results from the general property of the flexibility of dipeptides and protein that is primarily determined by the flexibility of the constituent amino acids [10]. In particular, glycyl-L-alanine dipeptide experiences torsional, orientational and displacive changes in different environments [7].

We report here the crystal structure of a synthesized polymorph, referred to as Gly-L-Ala.HI.H_2_O (Poly2), and its pyroelectric, piezoelectric and optical second harmonic generation properties. All these properties are allowed by symmetry in polar point group 2. In fact, the existence of these properties is determined, for any crystalline material, by its point group symmetry.

Second harmonic generation is a nonlinear optical property displayed by acentric crystals that results from the nonlinear behavior of the molecular electronic system to an intense external optical field. The property has revealed the non-centrosymmetric character of biological structures, from amino acids to dipeptides, proteins and also viruses [11,12,13].

Piezoelectricity is the ability of a crystalline material to generate an output voltage between two parallel faces as a response to an applied external force. Therefore, the property allows conversion between mechanical deformation and electricity arising from the absence of inversion symmetry in a crystalline structure. Consequently, mechanical energy can be harvested through the piezoelectric effect [14].

Electromechanical coupling is a phenomenon exhibited by amino acids [15,16,17], dipeptides such as chiral diphenylalanine nanotubes [18] and its derivatives [19,20] and cyclic dipeptides such as cyclo-L-phenylalanine-L-tryptophan [21], cyclo-glycine-L-tryptophan [22] and cyclo-L-tryptophan-L-tryptophan [23].

The piezoelectric behavior displayed by amino acids and dipeptides enables them to be viewed as potential materials to be integrated into energy harvesting devices: it has been reported that the metastable amino acid *β*-glycine embedded into electrospun polymer fibers displays enhanced piezoelectric and nonlinear optical properties [24]. Chiral diphenylalanine (PhePhe) nanotubes were reported to show a shear component of the piezoelectric tensor of 60 pm/V, and fabricated energy harvesters were able to generate voltage and power up to 2.8 V and 8.2 nW, respectively, with a 42 N force applied periodically [25,26]. Derivatives of PhePhe incorporated into electrospun fibers exhibit strong piezoelectric properties [19,20].

## 2. Experimental Section

### 2.1. Synthesis

Cyclo-glycyl-L-alanine (1.29 g, 10 mmol) was dissolved in 5 mL HI (57%, stabilized with H_3_PO_3_) and 10 mL of water. H_3_PO_3_ acted as a stabilizer for HI to avoid reduction to elemental iodine. After two weeks of slow evaporation at room temperature, transparent, hexagonal-shaped single crystals of Gly-L-Ala.HI.H_2_O (Poly2) were formed. The crystals were collected and rinsed with acetone, dried and kept in a dissector. An example of the crystals grown is shown in Figure 1. Cyclo-glycyl-L-alanine (cyclo-Gly-L-Ala) was purchased from Bachem AG (Bubendorf, Switzerland). Hydriodic acid (HI) was purchased from Merck (Darmstadt, Germany) and used as received.

The synthesis of Gly-L-Ala.HI.H_2_O (Poly2) started with the cyclic form of the dipeptide (cyclo-glycine-L-alanine) and therefore the crystal growth conditions are different from those reported for Gly-L-Ala.HI.H_2_O (Poly1). We have also attempted the synthesis of linear glycyl-L-alanine with HI in an aqueous solution as described for Gly-L-Ala.HI.H_2_O (Poly2) using cyclo-glycyl-L-alanine; however, no crystals were ever formed. After complete evaporation of the solution, an oily residue was obtained. We conclude that Gly-L-Ala.HI.H_2_O (Poly1) is only obtained following the procedure reported in reference [1].

### 2.2. X-ray Crystallography Experimental Conditions Description

A single crystal of Gly-L-Ala.HI.H_2_O (Poly2) was selected, covered with Fomblin (polyfluoro ether oil) and mounted on a nylon loop. Data were collected at 293(2)K on a Bruker D8 Venture diffractometer equipped with a Photon 100 CMOS detector, using graphite monochromated Mo-Kα radiation (λ = 0.71073 Å). The data was processed using the APEX3 suite software package, which includes integration and scaling (SAINT), absorption corrections (SADABS 2016/2) [27] and space group determination (XPREP). The structure solution and refinement were performed using direct methods with the programs SHELXT (version 2014/5) and SHELXL (version 2018/3) [28] contained in the APEX and WinGX-Version 2021.3 [29] software packages. All non-hydrogen atoms were refined anisotropically. Except for NH, OH and water H-atoms, which were located on the difference Fourier map, the remaining hydrogen atoms were inserted in idealized positions and allowed to refine riding on the parent carbon or oxygen atom with C–H distances of 0.96 Å, 0.97 Å and 0.98 Å for methyl, methylene and methine H atoms, respectively. The molecular diagrams were drawn with Mercury [30]. Crystal data for Gly-L-Ala.HI.H_2_O (Poly2) are presented in the Appendix A. The data for Gly-L-Ala.HI.H_2_O (Poly2) was deposited in the CCDC under deposit number 2247398.

### 2.3. Dielectric Spectroscopy

The dielectric properties of the Gly-L-Ala.HI.H_2_

O (Poly2) crystals were characterized by impedance spectroscopy at temperatures of 288–383 K and in the frequency range 20 Hz–3 MHz. The complex permittivity, written as ε = ε′ − iε″, where ε and ε″ are the real and imaginary parts, respectively, was calculated from the measured capacitance (C) and loss tangent (tan δ), using the equations:C = ε′ε_0_(A/d) and tan δ = ε″/ε′(1)

Here, A is the electric contact area and d is the crystal thickness. To form the capacitor, the bottom and top electrodes were gold contacts sputtered onto the sample surfaces. A Wayne Kerr 6440A (Wayne Kerr Electronics, London, UK) precision component analyzer was used, together with a dedicated computer and software, to acquire the data. Shielded test leads were employed to avoid parasitic impedances due to connecting cables. Temperature-dependent measurements were performed at a rate of 2 °C/min using a Polymer Labs PL706 PID controller (Polymer Labs, Los Angeles, CA, USA) and furnace.

### 2.4. Pyroelectric Measurements

Pyroelectricity is a property of polar crystalline materials that results from the temperature dependence of their spontaneous polarization. By changing the temperature, an electric field originating from the changes in intrinsic dipoles is compensated for by the surface layer of free charges [31]. The rate of change of the spontaneous polarization ($P_{s})$ with the temperature (T) is the pyroelectric coefficient, Ps=dPsdT. The change in polarization was detected by measuring, at constant stress, the pyroelectric current, I=A(dPsdT)(dTdt), with a Keithley 617 electrometer (Keithley Instruments GmbH, Landsberg, Germany). In the equation, A is the electrode area and dTdt is the rate of temperature change with time (t).

The measurements were performed on a capacitor geometry under short-circuit conditions and the electrode area was 8.11 × 10^−6^ m. The temperature interval was between 290 K and 345 K at a heating rate of 2 K/min.

### 2.5. Second Harmonic Generation

The second harmonic measurements were carried out using a mode-locked Ti:sapphire laser (model: Mira, Coherent Inc., Santa Clara, CA, USA) coupled into a Nikon (model: Eclipse Ti2, Nikon Europe B.V., Amstelveen, The Netherlands) inverted microscope, as shown in Figure 2. A calcite Glan–Taylor polarization followed by a zero-order half-wave plate controlled the incident polarization. A Nikon CFI Plan Fluor ×10 objective focused the beam onto the samples, while a Mad City Labs (Madison, WI, USA) xyz piezo-controlled translation stage positioned the samples in the focal plane with sub-micrometer accuracy. Incident powers ranged from 5 mW for the sample to approximately one hundred microWatts for the BBO crystal used for calibration of the system’s sensitivity. Although the Fourier limit of the pulse duration is approximately 85 fs, we estimate that the duration stretched to approximately 120 fs when incident on the sample because of the combined effect of the calcite polarizer and the microscope objective. The detection arm along the transillumination direction consisted of a 40 mm focal length best form lens (model:LBF254-040-A, Thorlabs, Newton, NJ, USA) to collimate the second harmonic light followed by a zero-order half-wave plate polarizer combination to analyze the emitted second harmonic light. A long-pass dichroic mirror (Thorlabs DMLP650) filters out most of the incident light while reflecting 98% of the second harmonic light. A short focal length lens then focuses the beam through a narrow band-pass filter (Thorlabs FBH400-40 nm) onto a fiber bundle coupled to an Andor imaging spectrometer (model: Shamrock 300i, Andor Technology Belfast, UKequipped with a cooled CCD array (model: Andor Technology Belfast, Newton, UK). 

A crystal plate taken from the crystal sample shown in Figure 1 was mounted perpendicular to the laser beam and the surface was scanned for the most intense SHG signal. We acquired the second harmonic signal using the following protocol. At each position of the half-wave plate controlling the polarization of the incident beam, the analyzer half-wave plate was scanned over 180°. At each analyzer wave plate position, the CCD signal was integrated for 1 s and the subsequent second harmonic light spectra were fitted to a Gaussian profile as shown in Appendix A. The area under the Gaussian fit was taken as the total number of second harmonic signal counts. 

### 2.6. Piezoelectric Measurements

The piezoelectric properties of Gly-L-Ala.HI.H_2_O (Poly2) were analyzed by embedding the crystals within fibers fabricated by a conventional electrospinning technique described previously [32]. To produce the fibers, a clear and homogeneous 10 % polymer solution was prepared by dissolving 0.5 g of poly (methyl methacrylate) (PMMA, Mw 996,000, Sigma-Aldrich, Schenlldorf, Germany) in 5 mL of chloroform. To this solution, 0.5 g of Gly-L-Ala.HI.H_2_O (Poly2) powder was added at a 1:1 weight ratio. The resulting mixture was stirred for several hours under ambient conditions before the electrospinning process. This precursor solution was loaded into a syringe and its needle was connected to the anode of a high-voltage power supply (model: CZE2000 Spellmann, Broomers Hill Park, UK). To produce in-plane fibers, the spinning voltage was set at 18 kV, with a distance of 12 cm between the anode and the collector. The flow rate of 0.10 mL/h was controlled by a syringe pump with an attached needle of 0.8 mm diameter. The fiber mat for piezoelectric measurements was collected on high-purity aluminum foil, which served as the electrodes.

The crystallinity and crystallographic orientation of Gly-L-Ala.HI.H_2_O (Poly2) inside the fibers were studied by XRD. The diffraction pattern was recorded between 5° and 50° using θ–2θ scans on a Philips (Amsterdam, The Netherlands) PW-1710 X-ray diffractometer with Cu-Kα radiation of wavelength 1.5406 Å. The morphology and fiber thickness were determined using a Nova NanoSEM scanning electron microscope operated at an accelerating voltage of 10 kV. The Gly-L-Ala.HI.H_2_O (Poly2)@PMMA microfibers were deposited on a silica surface previously covered with a thin film (10 nm thick) of Au–Pd (80–20 weight%) using a high-resolution sputter cover, 208 HR Cressington Company, coupled to an MTM-20 Cressington high-resolution thickness controller.

The piezoelectric output voltage and current were measured across a 100 MΩ load resistance connected to a low-pass filter, followed by a low-noise preamplifier (SR560, Stanford Research Systems, Stanford, CA, USA), before being recorded with a digital storage oscilloscope (Agilent Technologies DS0-X-3012A, Waldbronn, Germany). The fiber array sample with a (30 × 40) mm^2^ area (200 µm thickness) was subjected to applied periodic mechanical forces imposed by a vibration generator (model: SF2185, Frederiksen Scientific, Olgod, Denmark), with a frequency of 3 Hz determined by a signal generator (model: 33120A, Hewlett Packard, Palo Alto, CA, USA). The applied forces were calibrated using a force-sensing resistor (FSR402, Interlink Electronics Sensor Technology, Graefelfing, Germany). The sample was fixed on a stage, and the forces were applied uniformly and perpendicularly over the surface area.

## 3. Results and Discussion

### 3.1. Crystal Structure

Gly-L-Ala.HI.H_2_O (Poly2) crystallizes as yellowish prisms in the monoclinic system, space group P2_1_, as a glycine-L-alanine hydroiodide salt with one water molecule. Its molecular structure is depicted in Figure 3, and the most relevant bond distances and angles are given in the caption of the corresponding Figure and in Appendix A.

As referred to above, Gly-L-Ala.HI.H_2_O (Poly2) is a polymorph of Gly-L-Ala.HI.H_2_O (Poly1) that was reported by Kehrer et al. in 1989 [1]. However, no crystal data could be found in the Cambridge Crystallographic Data Centre (CCDC) [33] for graphical comparison. Both structures have comparable unit cells (a = 7.747(6) Å, b = 6.435(5) Å, c = 10.941(9) Å, α = 90°, β = 107.53(3)°, γ = 90° for Gly-L-Ala.HI.H_2_O (Poly2) vs. a = 10.933(3) Å, b = 6.371(2) Å, c = 7.709(1) Å, α = 90°, β = 107.29(1)°, γ = 90° for Gly-L-Ala.HI.H_2_O (Poly1) but with axis *a* and *c* interchanged. The unit cell atomic coordinates are different in both crystals, the more evident being the location of the iodide anion, which in Gly-L-Ala.HI.H_2_O (Poly1) has *y* fixed at 0.5b and in Gly-L-Ala.HI.H_2_O (Poly2) is at 0.6538b.

In Figure 4a,b, projections of the crystal structure down the crystallographic axes *a* and *c* are shown for Gly-L-Ala.HI.H_2_O (Poly2). Comparing these figures with those depicted in ref. (1), one can see that although the unit cell parameters are similar, the atom positions are different. The 2_1_ polar axis is parallel to the crystallographic b axis, as shown in Appendix A. If we consider a partial electrical dipole formed by C5-C3-C4, this dipole points in the 2_1_ polar axis direction with an inclination of around 30°, Figure 4a. Moreover, the dipeptide backbone N1-C1-C2-N2-C3 also forms a partial electrical dipole inclined also ca. 45° to the 2_1_ polar axis, Figure 4b. 

The ammonium group (N1) displays a slightly distorted tetrahedral geometry, with the N–H distances within the group varying between 0.885(3) and 0.880(3) Å. On the other hand, the carboxylate group shows two distinct C–O bond lengths (C4-O2 1.221(7) and C4-O3 1.316(7) Å), clearly indicating that the group is in the carboxylic acid form and allowing the assignment of the carbonyl and O–H substituents. 

The main chain of the molecule presents an almost planar *trans* conformation between atoms N1 and C3 (Appendix A), with torsion angles of 167.9(5)° for N1-C1-C2-N2 and 177.7(5)° for C1-C2-N2-C3. In Poly2, the angle for C4-C3-N2-C2 is 111.0(6)°. The torsion angle of −77.12° in C4-N1-C2-C1 from [9] indicates a gauche conformation.

The supramolecular arrangement observed in the crystal structure of Gly-L-Ala.HI.H_2_O (Poly2), when viewed along the b axis (ac plane), shows consecutive layers of iodide anions and peptide cations parallel to the b axis (Figure 5), with the peptide cations antiparallel.

This 3D arrangement is further stabilized by the presence of water molecules that interact with the peptide moieties through intermolecular hydrogen bonds within the asymmetric unit and with other symmetry-generated cations. Furthermore, all remaining hydrogen atoms bonded to electronegative O or N atoms participate in either classical (N–H…O and O–H…O) or nonclassical hydrogen bonds (N–H…I and C–H…I), as presented in Table 1. 

The iodide anion is involved in three H-bonds, one of them within the asymmetric unit. The same applies to the three ammonium H-atoms, which are involved in two N–H…I nonclassical H-bonds (H10A and H10C), whereas H10B participates in two classical H-bonds, with the water molecule (O4) and with the carbonyl (O2) of the carboxylate group. On the other hand, oxygen O1 simultaneously accepts two H-bonds from H40 and H41 of two water molecules.

The ammonium group (N1) displays a slightly distorted tetrahedral geometry, with the N–H distances within the group varying between 0.885(3) and 0.880(3) Å.

The faces of Gly-L-Ala.HI.H_2_O (Poly2) used for characterization studies and crystal structure determination, with the assigned Miller indices, are indicated in Figure 6a. The complete crystal morphology is depicted in Appendix A.

### 3.2. Pyroelectric Properties

The pyroelectric coefficient is a vector quantity with three components ($p_{1},p_{2},p_{3}).$ For space group P2_1_, with the 2-fold screw axis along the b-axis, the vector has only one component along that crystallographic axis. The pyroelectric coefficient reported in this work was measured on a (010) orientated Gly-L-Ala.HI.H_2_O (Poly2) crystal plate. Its value reached a maximum of $p=45\;\mu C/m^{2}K$ at 345 K, as shown in Figure 7. The pyroelectric coefficient along the polar 2-fold axis reported for the Gly-L-Ala.HI.H_2_O (Poly1) polymorph varied between $p=2\;\mu C/m^{2}K$ at 100 K and $p=15.5\;\mu C/m^{2}K$ at 357 K [2]. Therefore, the pyroelectric coefficient of Poly2 is roughly three times bigger than that reported for Poly1. This is due to the different crystal structure arrangements inside the crystalline unit cell.

There are in the crystal unit cell of Gly-L-Ala.HI.H_2_O (Poly2) four dipole moments: two from the water molecules and another two formed by ${NH}_{3}^{+}$…$I^{-}$, as shown in Figure 8. The first two dipoles form an angle of approximately 30° with the polar b-axis, whereas the other two dipoles ${(NH}_{3}^{+}$…$I^{-}$) form an angle of approximately 20°. For Gly-L-Ala.HI.H_2_O (Poly1), similar dipoles are identified from the reported structure. However, the water dipoles are inclined to the b-axis by 45° [2]. Additionally, for ${NH}_{3}^{+}$…$I^{-}$, the bond length in Poly1 (2.64Å) is shorter than the corresponding bond length in Gly-L-Ala.HI.H_2_O (Poly 2) (2.97Å). As a consequence, the overall net dipole moment contribution is higher for Poly2 than for Poly1, which explains the higher value for the pyroelectric coefficient reported in this work.

In Table 2, the pyroelectric coefficients of some important inorganic and semi-organic crystals are presented. 

### 3.3. Thermal Properties

TGA measurement results indicate that the crystal mass is stable up to 380 K. A small initial mass loss of 2% happens at 388 K that results from water molecule evaporation (SI, Appendix A), which is also visible in the DSC at 386 K (SI Appendix A). At around 393–413 K, the iodine hydrogen bonds break, visible at the sharp enthalpy peak of the DSC (393 K), alongside some mass loss. Above 473 K, there is a considerable loss of mass (about 50%), which corresponds to the decomposition of the crystalline compound. The peak at 533 K corresponds to the degradation temperature of the dipeptide glycyl-L-alanine. It is interesting to note that this temperature, 533 K, is very close to the melting temperature reported for cyclo-glycyl-L-alanine, which is 531 K, 32 K lower than that for the linear glycyl-L-alanine dipeptide (563 K) [39]. This suggests that although the dipeptide in Poly2 is not in its cyclic form when crystallized as the present hydroiodide salt, it keeps a memory of its initial cyclic closed chain, therefore showing a thermal memory effect (one should remember that the crystal synthesis started from the cyclic dipeptide form and after reacting with the iodide acid the cyclic chain opened up).

### 3.4. Dielectric Spectroscopy

Figure 9 and Figure 10 show the temperature dependence of the real (ε′) and imaginary (ε″) parts of the dielectric permittivity measured in the range 287 K–370 K for different frequencies from 100 Hz to 100 kHz. Two regimes are identified in both the real and imaginary parts of the dielectric permittivity, with different dependences below and above 350 K. The real part of the electric permittivity is approximately constant and less than 100 from room temperature until 343 K, both for low and high frequencies. However, beyond 350 K it increases very steeply, reaching the value of 1100 for 100 Hz, as shown in Figure 9.

Similarly, the imaginary part of the electric permittivity, ε″, is also approximately constant until 350 K and smaller than 25 for all frequencies, as seen in Figure 10. Again, it increases steeply beyond 350 K, reaching 160 000 at T ~ 375 K and a frequency of 100 Hz.

Figure 11 and Figure 12 show the frequency dependence of the real and imaginary electric permittivity at different temperatures. Both ε′ and ε″ present an initial sharp drop in the low-frequency region and afterwards attain a slower decrease at high frequencies. The initial, low-frequency drop in the imaginary part reveals a contribution from a conductivity term, as the samples are non-ideal capacitors.

For purely electronic conductivity, the permittivity is imaginary and given by ε″ = $\sigma_{DC}$/($\varepsilon_{0}$ω) [40], where $\varepsilon_{0}$ is the vacuum dielectric permittivity, ω is the angular frequency and $\sigma_{DC}$ is the DC conductivity. For ionic charge carriers that cause electrode or Maxwell–Wagner polarization effects, this equation can be generalized, so that the conductivity contribution can be described by the equation ε″ = $\sigma_{DC}$/($\varepsilon_{0}$ω^s^), where s is an exponent and s ≤ 1. As such, since the logarithm of the imaginary component of the permittivity as a function of the logarithm of the frequency gives a linear dependence, it was fitted with a straight line to determine the conductivity according to: (2)$$ln(\varepsilon^{\prime \prime })=ln\left(\frac{\sigma_{DC}}{\varepsilon_{0}{\left(2\pi \right)}^{s}}\right)-s\;ln(f)$$

The inset of Figure 12 shows the linear fit to $ln(\varepsilon^{\prime \prime })$ as a function of the logarithm of the frequency for different temperatures in the low-frequency region. From the fits, the corresponding calculated DC conductivity ($\sigma_{DC}$) values and their temperature dependence are shown in Figure 13. Again, a small variation in the DC conductivity is observed until 350 K, above which the conductivity rises sharply. The $\sigma_{DC}$ behavior as a function of temperature in both regions shows characteristic Arrhenius-like processes, with activation energies (E_at_) given by the equation [40,41]:(3)$$\sigma =\frac{\sigma_{0}}{T}e^{-\frac{E_{at}}{k_{B}T}}$$
where T is the temperature, k_B_ is the Boltzmann constant and σ_0_ is a constant. The activation energy can be determined from the slopes of the fittings to the curves of ln⁡(σT) as a function of the inverse of T for the different temperature regions, as shown in the inset of Figure 13. The two temperature regions with different conductivities and the corresponding activation energy ($E_{at}$) values are shown in the figure. $E_{at}=0.07\;eV$ corresponds to the low-temperature region and is characteristic of electrical conduction through the polaron transport behavior [42]. At higher temperatures, the activation energy increases to $E_{at}=3.6\;eV$, which is characteristic of ionic conduction in the samples [40]. As such, the two observed regimes are due to the change from the low-temperature, polaronic transport behavior to the high-temperature ionic conductivity dependence. This region is associated with the onset of temperature-induced changes in the samples (e.g., loss of water as observed from the TGA results, which starts just above 350 K).

### 3.5. Second Harmonic Response

Data for the second harmonic response of an approximately 3 mm-thick Gly-L-Ala.HI.H_2_O (Poly2) dipeptide crystal (taken from that shown in Figure 1) were acquired using incident fundamental pulses with an average incident power of 5 mW, corresponding to roughly 66 pJ of energy per pulse. The beam was incident normal to the as-grown crystal surface, and the crystal was scanned over the laboratory x, y and z directions, with z taken to be the direction of the laser beam propagation. The maximum signal values as a function of the polarization of the fundamental beam direction are presented in Figure 14.

Second harmonic signals were observable only for a narrow range of z positions when the fundamental beam waist was within a few tens of μms from the crystal surface. Furthermore, the orientation of the analyzer that resulted in the maximum detected signal was very nearly parallel to the direction of the incident polarization. We believe this to be an indication that the normal of the as-grown crystal is close to the crystallographic b axis. As explained in the Appendix A, for this orientation, the second harmonic light will be generated with nearly the same polarization as the fundamental beam and will suffer from strong phase mismatch, limiting the generation to close to the crystal surface. 

We have carried out a study of the second harmonic response as a function of the fundamental beam’s waist position by translating the crystal in 10 μm steps along the beam’s propagation direction using the MadCity’s piezoelectric translation stage. Representative data are shown in Figure 15, along with a theoretical fit as described in the Appendix A.

By calibrating the efficiency of our second harmonic microscope using a 2 mm-thick BBO crystal cut, the phase matching angle for 800 nm incident light, we can estimate a lower bound for the effective nonlinear susceptibility of the dipeptide crystal. Applying the results described in the Appendix A, we estimate that a lower bound for the effective second-order nonlinear coefficient of Gly-L-Ala.HI.H_2_O (Poly2) crystals is $d_{eff}\geq$0.14 pm/V. 

### 3.6. Piezoelectric Response

Any pyroelectric material is, by symmetry, also a piezoelectric and nonlinear optical material. Additionally, the tensor describing piezoelectricity and SHG properties of crystalline materials are the same (in this case, that for point group 2), as presented in S6. Therefore, the suitability of Gly-L-Ala.HI.H_2_O (Poly2) crystals to be used as piezoelectric nanogenerators for energy harvestings was investigated by embedding them into electrospun nanofibers as described before.

An interconversion between a mechanical and an electrical stimulus arising due to applied uniform stress, which generates electric polarization inside a dielectric material, is the origin of the piezoelectric effect. For a crystalline solid to display this phenomenon, it must have a crystal structure without inversion symmetry. Gly-L-Ala.HI.H_2_O (Poly2) crystallizes in the polar point group 2, which is acentric. The tensor relationship between the polarization $P_{j}$ and stress $\sigma_{k}$, $P_{j}=d_{jk}\sigma_{k}$ is given by the piezoelectric coefficient $d_{jk}$ [43]. There is no preferential crystallographic orientation of the compound inside the electrospun fibers (see Appendix A). Therefore, a polarization develops under forces applied repeatedly at regular times perpendicularly to the fiber array, and an effective piezoelectric modulus $d_{eff}$ is measured along the same direction.

The applied stress (force per unit area) ranged between $1.5\times 10^{2}\;Nm^{-2}$ and $8.0\times 10^{2}\;Nm^{-2}$. For Gly-L-Ala.HI.H_2_O (Poly2)@PMMA fiber mats, a 1.0 N applied periodical force gave rise to a maximum instantaneous output piezoelectric voltage and current of 28 V and 280 nA, respectively, as shown in Figure 16a. Here, the two opposite peaks correspond to the press and release of the fiber mat. A plot of the output voltage as a function of several applied periodic forces shows an output voltage increasing linearly with the force magnitude as expected (Figure 16b). The PMMA polymer matrix is not piezoelectric and does not contribute to the measured piezoelectric voltage. Taking into account a response time of 1 ms, the magnitude of $d_{eff}$ is obtained from the integration of the induced piezoelectric current over that period of time, Q=∫Idt, resulting in Q=280 pC for Gly-L-Ala.HI.H_2_O (Poly2)@PMMA fiber mats. This induced charge, which is related to the applied force by the equation $Q=F\;d_{eff}$, allows us to calculate an effective piezoelectric coefficient equal to $d_{eff}=280\;pCN^{-1}$. This value is of the same order of magnitude as that obtained for organic–inorganic ferroelectric perovskite trimethylchloromethyl ammonium trichloromanganese ((TMCM)MnCl_3_), where $d_{33}=185\;pCN^{-1}$, and barium titanate (BaTiO_3_), with $d_{33}=190\;pCN^{-1}$ [44,45]. It is also important to compare the present result, $d_{eff}=280\;pCN^{-1}$, with that obtained for lead-free organic–inorganic perovskite (N-methyl-N′-diazabicyclo [2.2.2]octonium)–ammonium triiodide (MDABCO-NH_4_I_3_) embedded into PMMA electrospun fibers (MDABCO-NH_4_I_3_@PMMA in a 1:5 ratio), which was reported to be $d_{eff}=64\;pCN^{-1}$ [46], (Table 3).

It is remarkable that this new organic–inorganic Gly-L-Ala.HI.H_2_O (Poly2) crystalline compound exhibits, when embedded into electrospun fibers, a very highly effective piezoelectric coefficient that is similar in magnitude to an organic–inorganic perovskite also containing the iodide ion. In the present work, we demonstrate that Gly-L-Ala.HI.H_2_O (Poly2)@PMMA fibers may be incorporated into nanogenerators as active piezoelectric materials.

## 4. Conclusions

A polymorph (Gly-L-Ala.HI.H_2_O (Poly2)) of a previously reported glycyl-L-alanine HI.H_2_O salt was synthesized from the chiral cyclo-glycyl-L-alanine dipeptide. The dipeptide is known to show molecular flexibility in different environments, which originated the polymorphism. The crystal structure of the glycyl-L-alanine HI.H_2_O polymorph is determined at room temperature in space group P2_1_; therefore, it is a pyroelectric, piezoelectric and nonlinear optical material.

The pyroelectric coefficient reported in this work on a (010)-orientated Gly-L-Ala.HI.H_2_O (Poly2) crystal plate showed an increase with temperature with no significant abnormalities in the range 300–345 K, reaching a maximum of $p=45\;\mu C/m^{2}K$ at 345 K. Therefore, the pyroelectric coefficient of Poly2 is roughly three times higher than that reported for the Gly-L-Ala.HI.H_2_O (Poly1) polymorph, which was $p=15.5\;\mu C/m^{2}K$ at 357 K. The different orders of magnitude of the measured values for the two polymorphs results from the different atomic coordinates of the dipeptide, water molecules and the iodine ions within the unit cells of both compounds, which creates differences in some bond lengths and the directionality of hydrogen bonds.

Thermal studies showed that Gly-L-Ala.HI.H_2_O (Poly2) begins degradation at 533 K, close to the melting temperature reported for cyclo-glycyl-L-alanine, which is 531 K. That temperature (533 K) is 30 K lower than that reported for linear glycyl-L-alanine dipeptide (563 K), suggesting that although the dipeptide when crystallized in Poly2 is not in its cyclic form anymore, it keeps a memory of its initial closed chain, therefore showing a thermal memory effect.

The DC conductivity behavior as a function of temperature in the regions below and above 350 K indicates characteristics of electrical conduction through the polaron transport behavior in the low-temperature region and at higher temperatures, which is characteristic of ionic conduction in the samples.

The second harmonic generation efficiency of Gly-L-Ala.HI.H_2_O (Poly2) was measured against a state-of-the-art nonlinear optical barium borate (BBO) crystal. A lower bound for the effective second-order nonlinear coefficient of Gly-L-Ala.HI.H_2_O (Poly2) crystals was estimated $d_{eff}\geq$ 0.14 pm/V.

Finally, an effective piezoelectric coefficient equal to $d_{eff}=280\;pCN^{-1}$ was measured on an electrospun polymer fiber mat, Gly-L-Ala.HI.H_2_O (Poly2)@PMMA, demonstrating that the fibers are piezoelectrically active systems with great potential to be incorporated into energy harvesting devices.

## Figures and Tables

**Figure 1 materials-16-03690-f001:**
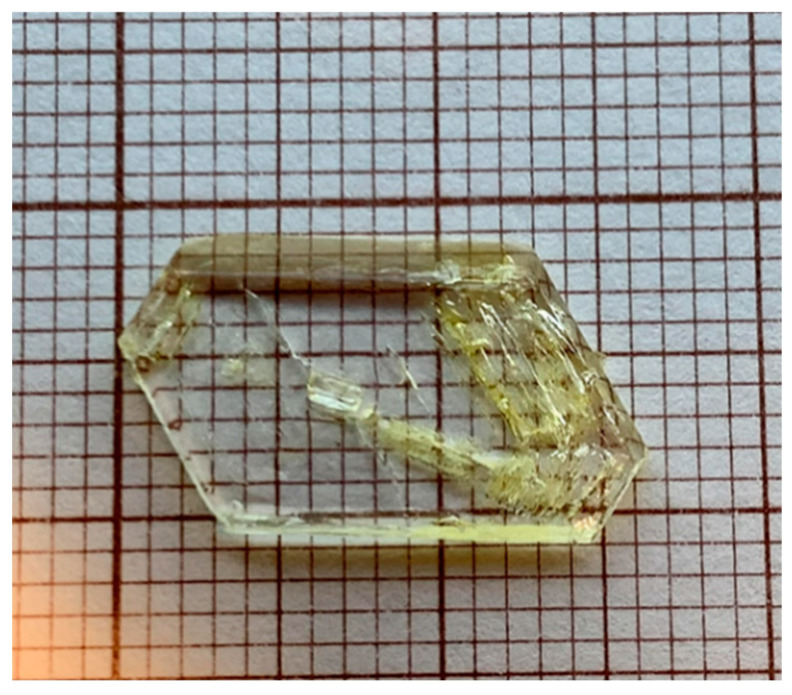
The glycyl-l-alanine hydroiodide monohydrate polymorph single crystal used for crystal structure determination and further characterization.

**Figure 2 materials-16-03690-f002:**
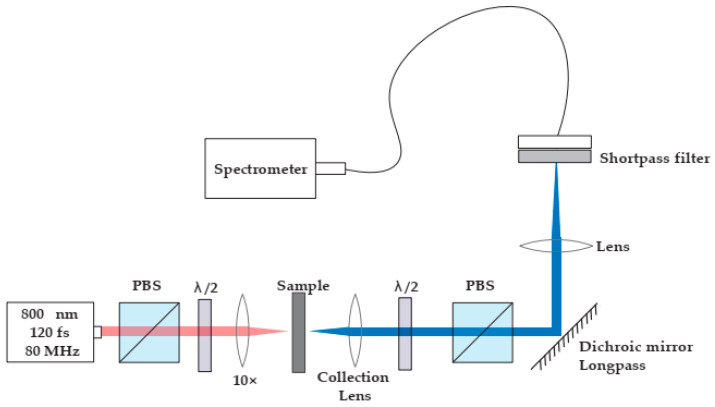
Second harmonic microscope layout; PBS—Polarized beam splitter; λ/2—half-wave plate. The transmission axes of both PBS are aligned vertically in this schematic.

**Figure 3 materials-16-03690-f003:**
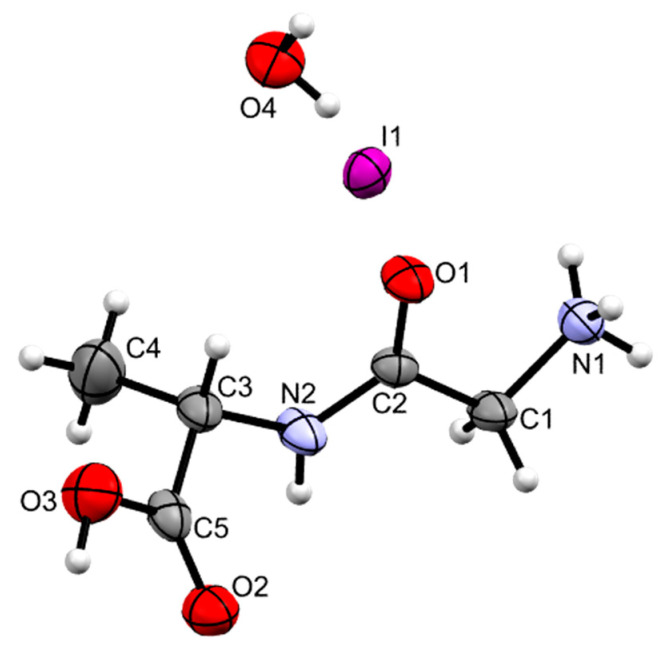
ORTEP representation of Gly-L-Ala.HI.H_2_O (Poly2), using 50% level ellipsoids. Selected bond distances (Å): C1-N1 1.481(4), C2-O1 1.243(6), C5-O2 1.221(7), C5-O3 1.316(7). Selected bond angles (°): C1-C2-N2 114.6(4), C2-N2-C3 124.4(4), N2-C3-C5 109.7(5), O2-C5-O3 123.9(5). Selected torsion angles (°): N1-C1-C2-N2 167.9(5), C1-C2-N2-C3 177.7(5).

**Figure 4 materials-16-03690-f004:**
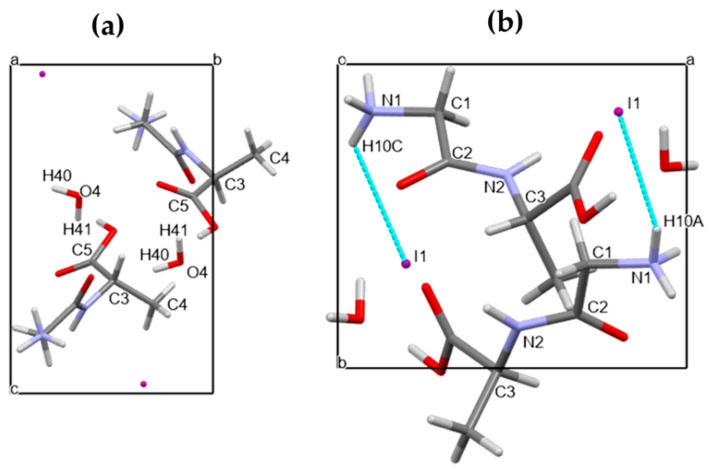
The unit cell of Gly-L-Ala.HI.H_2_O (Poly2) viewed down (**a**) the crystallographic a axis and showing the partial electrical dipole formed by C5-C3-C4, and (**b**) the crystallographic c axis showing the dipeptide backbone N1-C1-C2-N2-C3. Here a, b and c indicate the crystallographic axes.

**Figure 5 materials-16-03690-f005:**
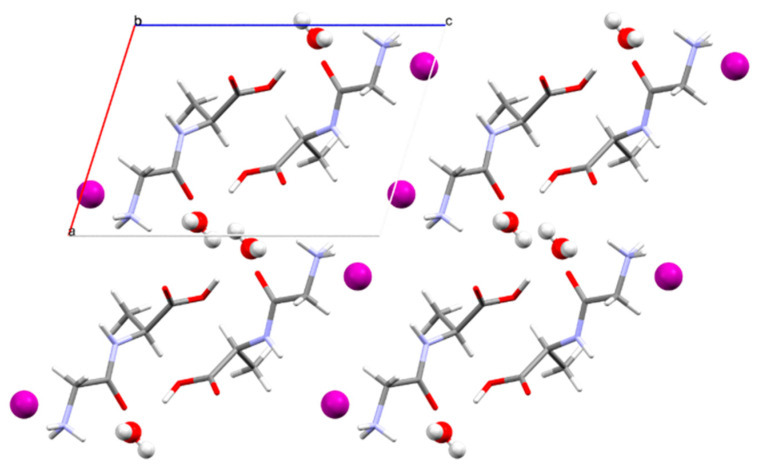
Supramolecular arrangement in Gly-L-Ala.HI.H_2_O (Poly2) viewed along the b axis (ac plane). Iodide anions and water molecules are represented in ball and stick style, whereas dipeptide cations are depicted in capped sticks style. Here a, b and c indicate the crystallographic axes.

**Figure 6 materials-16-03690-f006:**
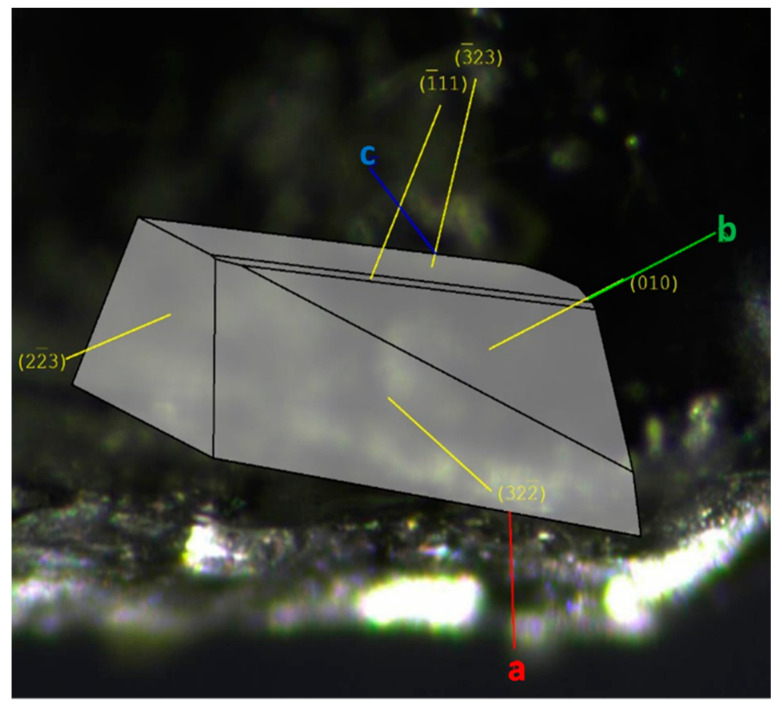
Morphology and face indexing of Gly-L-Ala.HI.H_2_O (Poly2) with axes a (red), b (green) and c (blue) and overlaid assigned Miller indices.

**Figure 7 materials-16-03690-f007:**
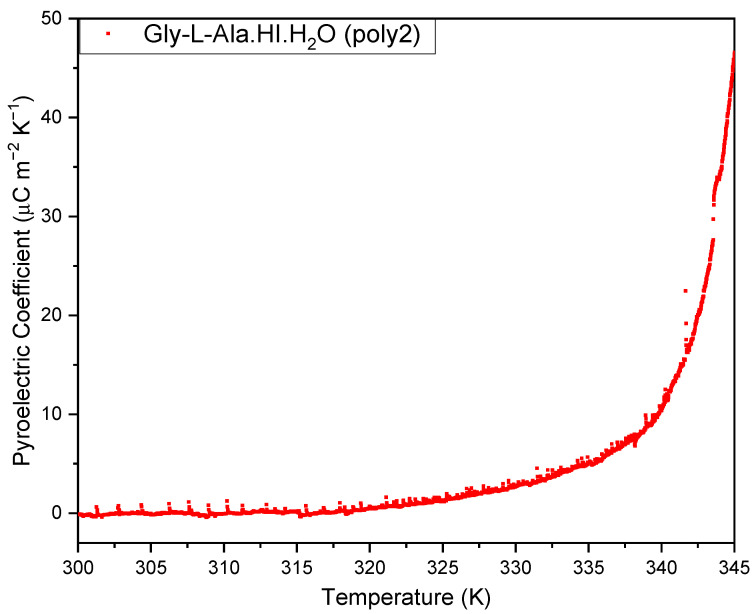
The pyroelectric coefficient versus temperature, measured during heating of the glycyl-L-alanine hydroiodide polymorph (Gly-L-Ala.HI.H_2_O (Poly2)) crystal from room temperature to 345 K. A maximum modulus of 45 µC/m^2^K occurs at 345 K.

**Figure 8 materials-16-03690-f008:**
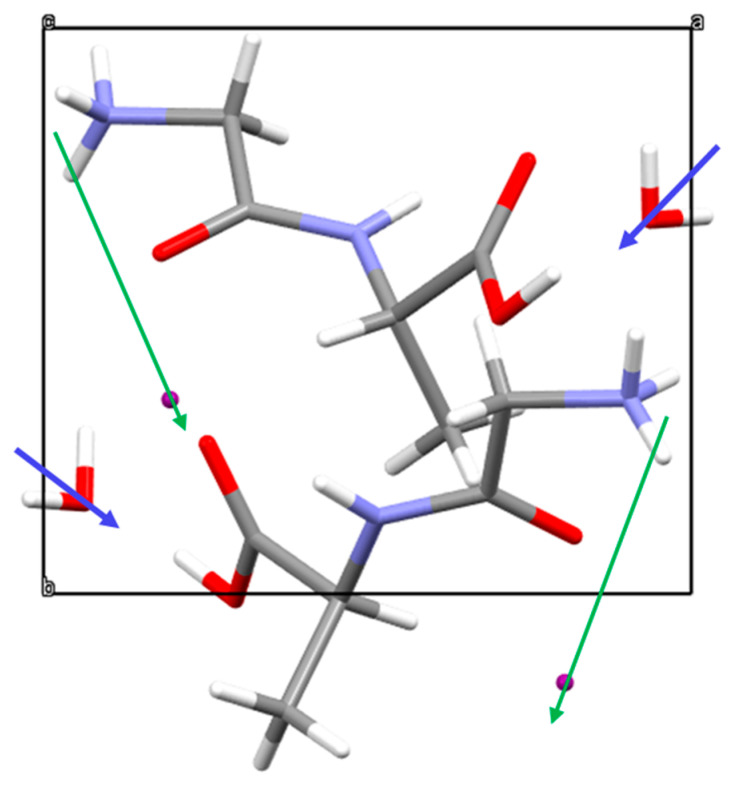
Gly-L-Ala.HI.H_2_O (Poly2) unit cell showing the electric dipole moments of the water molecule (blue arrows) and from the ${NH}_{3}^{+}$ group and $I^{-}$ ion (green arrows), indicating that there is a net dipole moment along the polar 2-fold screw b-axis.

**Figure 9 materials-16-03690-f009:**
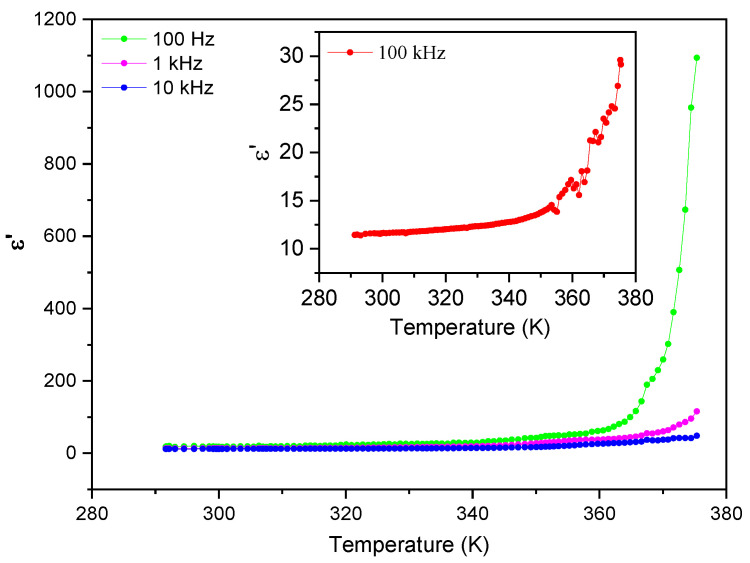
The real part of the electric permittivity, ε′, as a function of temperature for frequencies up to 10 kHz.

**Figure 10 materials-16-03690-f010:**
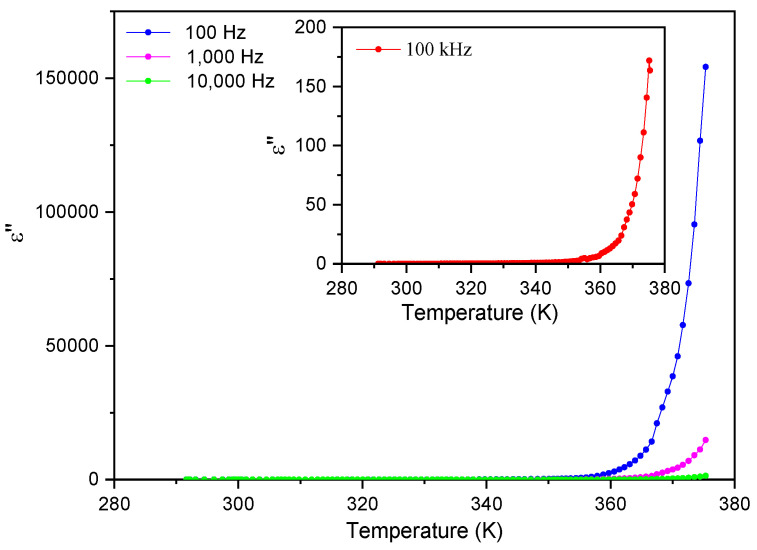
The imaginary part of the electric permittivity, ε″, as a function of temperature for frequencies up to 10 kHz.

**Figure 11 materials-16-03690-f011:**
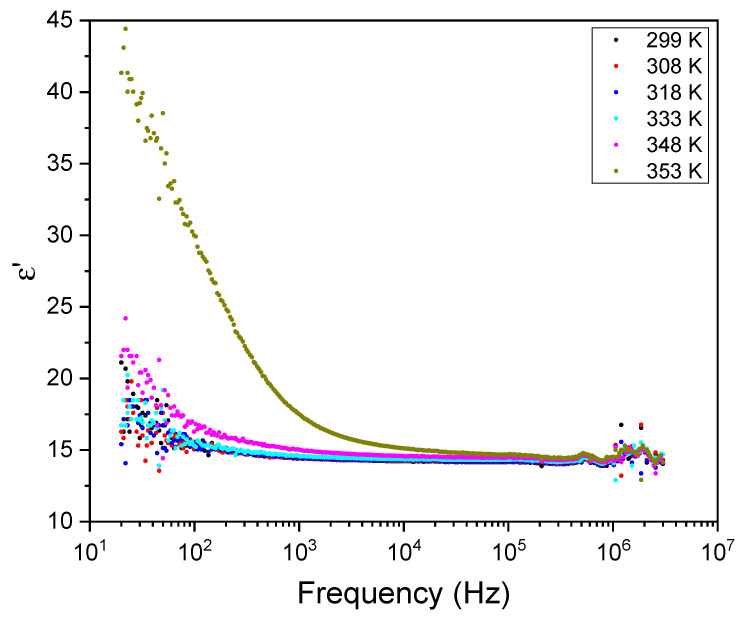
The real part of the electric permittivity, ε′, as a function of frequency for different temperatures.

**Figure 12 materials-16-03690-f012:**
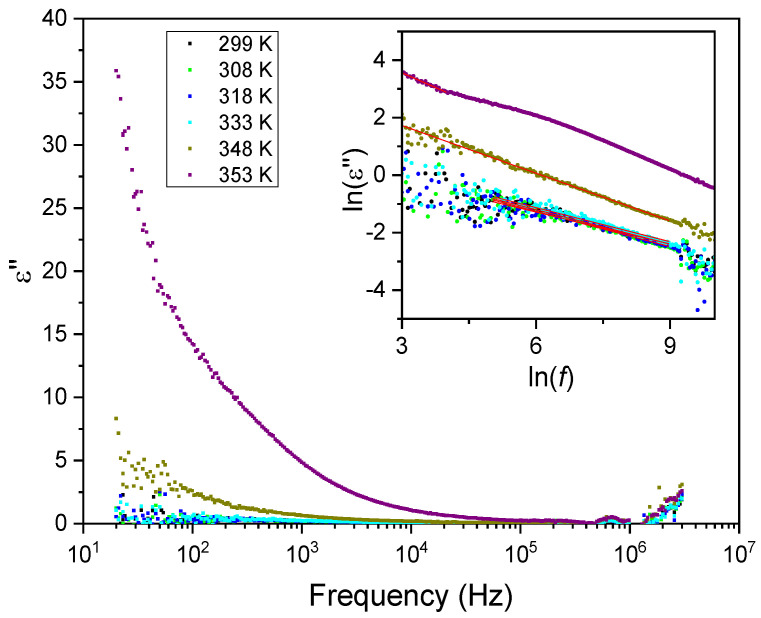
The imaginary part of the electric permittivity, ε″, as a function of frequency for different temperatures. The inset shows an ln–ln plot, where the linear dependence is observed due to the contribution from the DC conductivity (red lines).

**Figure 13 materials-16-03690-f013:**
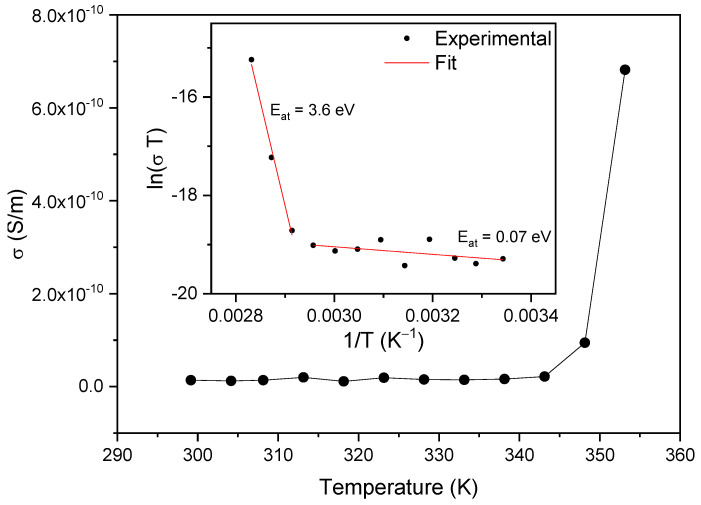
Temperature dependence of the electrical conductivity as determined from the low-frequency behavior of the imaginary permittivity of Figure 7. The inset shows $ln\left(\sigma T\right)$ as a function of the inverse of temperature. The slopes of the linear fits give the activation energies in the two temperature intervals.

**Figure 14 materials-16-03690-f014:**
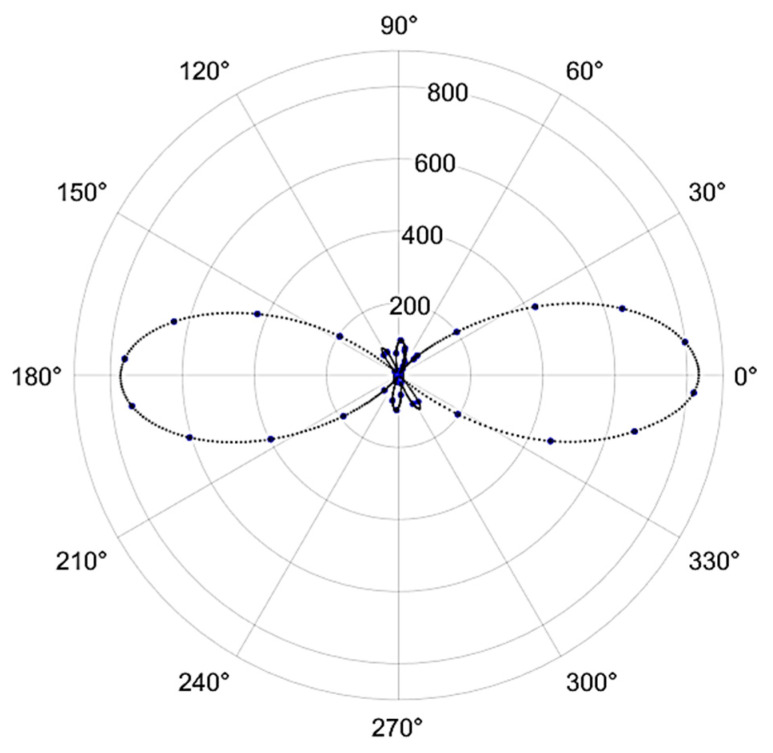
The second harmonic signal generated by the Gly-L-Ala.HI.H_2_O (Poly2) crystal as a function of the orientation of the fundamental beam’s polarization orientation relative to the transmission axis of the analyzer. Experimental points are in blue, whereas the dotted black line is a spline fit to the data.

**Figure 15 materials-16-03690-f015:**
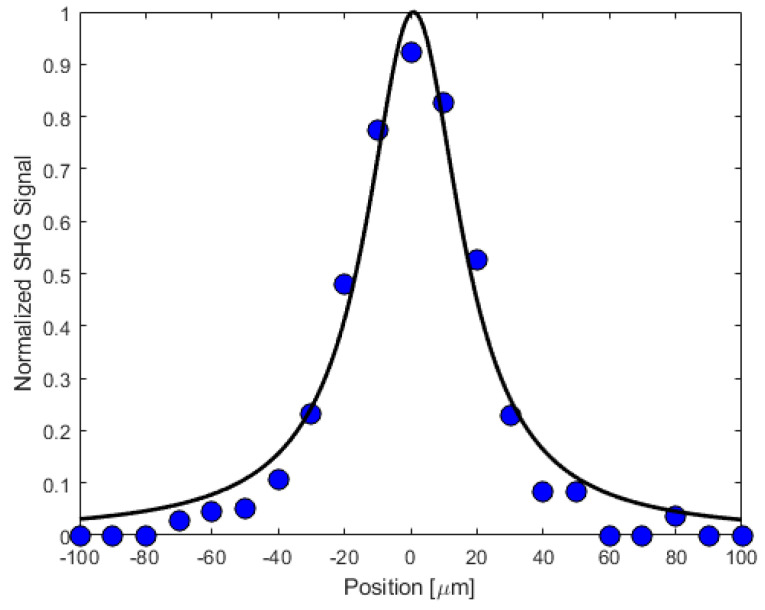
The second harmonic response (blue dots) as the crystal was translated in 10 μm steps along the propagation direction of the fundamental beam. Also shown is a theoretical fit (solid line) assuming Gaussian spatial and temporal profiles for the two beams. The 0 position was chosen to be close to the maximum SHG signal. We estimate that the coherence length due to phase mismatch is approximately 1.7 μm, whereas the Rayleigh range for the fundamental beam is roughly an order of magnitude larger at 16.4 μm.

**Figure 16 materials-16-03690-f016:**
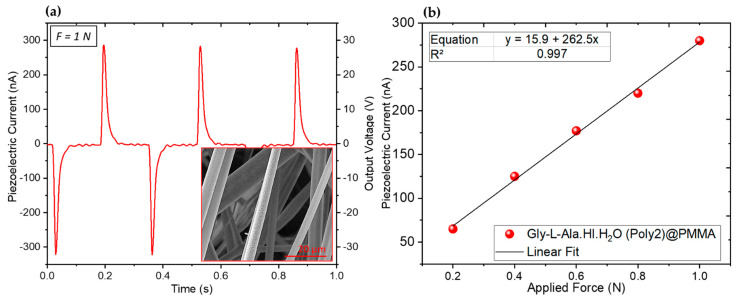
(**a**) Piezoelectric current and output voltage as a function of time, with an applied force of 1N and a frequency of 3 Hz. (**b**) Piezoelectric current as a function of different periodic forces applied from Gly-L-Ala.HI.H_2_O (Poly2) incorporated into PMMA polymer microfibers.

**Table 1 materials-16-03690-t001:** Hydrogen bonds for Gly-L-Ala.HI.H_2_O (Poly2) (Å and °).

D-H...A	d(H...A)	d(D...A)	<(DHA)	Symmetry Operation
C(1)-H(1B)...I(1)	3.26	3.953(3)	129.5	−*x* + 1, *y* − 1/2, −*z* + 2
N(1)-H(10B)...O(2)	2.30(6)	2.902(6)	128(6)	*x* − 1, *y*, *z*
N(1)-H(10B)...O(4)	2.47(6)	3.110(8)	133(6)	*x*, *y* − 1, *z*
N(1)-H(10A)...I(1)	2.77(4)	3.570(9)	152(6)	*x*, *y* − 1, *z*
N(1)-H(10C)...I(1)	3.13(11)	3.642(4)	119(10)	−*x*, *y* − 1/2, −*z* + 2
N(1)-H(10C)...I(1)	2.96(11)	3.598(9)	130(11)	
N(2)-H(20)...I(1)	2.76(10)	3.637(5)	169(7)	−*x* + 1, *y* − 1/2, −*z* + 2
O(3)-H(30)...O(4)	1.77(10)	2.662(6)	167(9)	−*x* + 1, *y* − 1/2, −*z* + 1
O(4)-H(41)...O(1)	2.026	2.824(4)	161.29(15)	−*x*, *y* + 1/2, −*z* + 1
O(4)-H(40)...O(1)	2.233	2.982(6)	152.86	

**Table 2 materials-16-03690-t002:** Characteristics of some semi-organic and organic materials compared with those of Gly-L-Ala.HI.H_2_O (Poly2) at room temperature. ε′ is the dielectric constant and *p* the pyroelectric coefficient.

Material	ε′	*p* (µC/m^2^K)	Ref.
Gly-L-Ala.HI.H_2_O (Poly2)	11.5	45 (345 K)	This work
Gly-L-Ala.HI.H_2_O (Poly1)	__	15.5 (357 K)	[2]
CsH(C_4_H_4_O_5_)∙H_2_O	11.2	2.5 (245 K)	[34]
LiH_3_(C_4_H_4_O_5_)_2_	8.1	6.6 (320K)	[35]
(NH_2_CH_2_COOH)_3_H_2_SO_4_	25	306 (319 K)	[36]
DL-Alanine	__	5.5 (305 K)	[37]
KH_2_PO_4_	45	300 (120 K)	[38]

**Table 3 materials-16-03690-t003:** Piezoelectric nanogenerator parameters for some cyclic dipeptides and a semi-organic perovskite crystal.

Active Compound	Force/Area (N/m^2^)	V_out_ (V)	d_eff_ (pC/N)	Power Density (μWcm^−2^)	Ref.
Gly-L-Ala.HI.H_2_O (Poly2)@PMMA(1:1) (Fiber mat)	8 × 10^2^	28	280	0.65	This work
Cyclo(L-Trp-L-Trp)@PCL (1:5) (Fiber mat)	5 × 10^3^	9.6	30	0.13	[23]
Cyclo(GW) (Crystal powder)	7 × 10^5^	1.2	5.6 *	0.002	[22]
Cyclo(FW) (Crystal powder)	6 × 10^5^	1.4	16 **	0.003	[21]
MDABCO-NH_4_I_3_@PMMA (1:5) (Fiber mat)	11 × 10^3^	6.1	64	0.09	[46]

* Calculated average piezoelectric coefficient from [22]. ** Calculated from data available in [21], assuming a nanogenerator time response of 0.5 s.

## Data Availability

Data is contained within the article or Appendix A and in the CCDC under deposit number 2247398.

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
