# Peer review of "A Polymorph of Dipeptide Halide Glycyl-L-Alanine Hydroiodide Monohydrate: Crystal Structure, Optical Second Harmonic Generation, Piezoelectricity and Pyroelectricity"

_materials, 2023, doi:10.3390/ma16103690_

Round 1
Reviewer 1 Report
Manuscript Number: materials-2306005
"A polymorph of dipeptide halide Glycyl-L-Alanine Iodide Hydrate: crystal structure, optical second harmonic generation and pyroelectricity"
In this paper, Baptista et al. studied the optical and pyroelectricity properties of the dipeptide-based polymorph system at various temperatures. Broadly speaking, the work is very clear and very informative for the development of quartz as an electronic material.
Authors have thoroughly studied the properties of the material and seem to be specialized in this field.
A few points that need to focus on are:
The introduction section needs to be revised in terms of materials properties, and optical second harmonic generation based.
The authors should provide the table of crystals refinement data rather than as a paragraph.

Author Response
Reviewer 1 Responses
The authors would like very much to thank the Reviewers for their valuable comments
- Reviewer: The introduction section needs to be revised in terms of materials properties, and optical second harmonic generation based.
Authors response: The introduction has been revised and information about optical second harmonic generation and piezoelectricity added. This text was added.
We report here the crystal structure of a synthesized polymorph, referred as Gly-L-Ala.HI.H2O (Poly2), its pyroelectric, piezoelectric and optical second harmonic generation properties. All these properties are allowed by symmetry in polar point group 2. In fact, these properties are determined, for any crystalline material, by its point group symmetry.
Second harmonic generation is a nonlinear optical property displayed by acentric crystals, resulting from the nonlinear behavior of the molecular electronic system to an intense external optical field. The property has revealed the non-centrosymmetric character of biological structures, from amino acids to dipeptides, proteins and also viruses [11-13].
Piezoelectricity is the ability of a crystalline material to generate an output voltage, between two parallel faces, as a response to an applied external force. The property therefore allows conversion between a mechanical deformation and electricity arising from the absence of inversion symmetry in a crystalline structure. Therefore, mechanical energy may be harvested through the piezoelectric effect [14].
Electromechanical coupling is a phenomenon exhibited by amino acids [15-17], dipeptides such as chiral diphenylalanine nanotubes [18] and its derivatives [19,20] and cyclic dipeptides like cyclo-L-phenylalanine-L-tryptophan [21], cyclo-glycine-L-tryptophan [22] and cyclo-L-tryptophan-L-tryptophan [23].
The piezoelectric behavior displayed by amino-acids and dipeptides, enabled them to be viewed as potential materials to be integrated in energy harvesting devices: It has been reported that metastable amino acid β-glycine embedded into electrospun polymer fibers, displays enhanced piezoelectric and nonlinear optical properties [24]. Chiral diphenylalanine (PhePhe) nanotubes, were reported to show a shear component of the piezoelectric tensor of 60 pm/V, and fabricated energy harvesters were able to generate voltage and power up to 2.8 V and 8.2 nW, respectively, upon a 42 N periodically applied force [25,26]. Derivatives of PhePhe incorporated into electrospun fibers exhibit strong piezoelectric properties [19,20].
- Reviewer:The authors should provide the table of crystals refinement data rather than as a paragraph.
Authors response: As written at the end of section 2.2, the crystallographic data for the crystal Gly-L-Ala.HI.H2O (Poly2), was deposited in the CCDC under the deposit number 2247398. All the details concerning the crystal structure determination and refinement can be retrieved freely from CCDC. A table with refinement details was added to supplementary information,Table SI3. The existing paragraph in section 2.2, which contained the summary of crystal data information for Gly-L-Ala.HI.H2O (Poly 2), has been deleted.
Table SI3: Crystal data and structure refinement for Gly-L-Ala.HI.H2O (Poly 2).
|
|
Gly-L-Ala.HI.H2O (Poly 2) |
|
Formula |
C5H13IN2O4 |
|
M |
292.07 |
|
l (Å) |
0.71073 |
|
T (K) |
293(2) |
|
crystal system |
Monoclinic |
|
space group |
P21 |
|
Crystal description |
Prism |
|
Crystal color |
Yellow |
|
a (Å) |
7.747(6) |
|
b (Å) |
6.435(5) |
|
c (Å) |
10.941(9) |
|
a (deg) |
90 |
|
b (deg) |
107.53(3) |
|
g (deg) |
90 |
|
V (Å3) |
520.1(7) |
|
Z |
2 |
|
rcalc (g cm-3) |
1.865 |
|
µ (mm-1) |
3.063 |
|
qmax (deg) |
28.284 |
|
total data |
12554 |
|
unique data |
2562 |
|
Rint |
0.0551 |
|
R [I>3s(I)] |
0.0286 |
|
wR2 |
0.0683 |
|
Goodness of fit |
1.097 |
|
r min |
-1.752 0.695 |

Reviewer 2 Report
The authors report the crystal structure, optical second harmonic generation and pyroelectric coefficient of dipeptide halide Glycyl-L-Alanine Iodide 2 Hydrate. There are a few comments listed below need to be addressed.
1. It seems like that the authors might mix the Experimental Section and Results and Discussion. I personally find that the contents of these two sections are a little bit similar. Experimental Section should include all the details about experiments and characterizations, other than results, analysis and discussion. Results and Discussion should focus on the analysis and discussion of obtained experimental results, not the details about the experiments.
2. For the crystal structure as shown in Figure 3, 4, 5, 8, how do the author obtain these results? How to estimate the accuracy?
3. From the results shown in Figure 7, the pyroelectric coefficient at room temperature is very small. Why? How to use this material as a pyroelectric or piezoelectric nanogenerator if the pyroelectric coefficient is so small?
4. As this material is proved to be pyroelectric, then it is must a piezoelectric material. Could the authors provide characterizations about its piezoelectricity?
5. Though the electrical conductivity increases a little when the temperature increases to over 350 K (but below 360 K), the conductivity value is still very small. Why? Would it have any influences on the performances of nanogenerators?
6. The authors present the optical second harmonic generation of this material. What is the relation of this property when using this material in nanogenerators? Since this manuscript is submitted to Special Issue: Piezoelectric Energy Harvesting and Sensing Technology: Materials, Mechanisms, and Applications, the authors are required to provide some related discussion about their proposed material’s possible application in piezoelectric energy harvesting and sensing.
7. The Conclusion Section is too long. The authors need to summarize the content.
8. Overall, the organization and English expression of the manuscript need to be improved significantly. Please also carefully check out the whole manuscript to correct typos and grammar issues.
According to the comments as listed above, I recommend a major revision decision to the manuscript. Publication in Materials could be considered only if the authors carefully and properly address the above listed comments.
Author Response
Reviewer 2 Responses
The authors would like very much to thank the Reviewers for their valuable comments
- Reviewer: It seems like that the authors might mix the Experimental Section and Results and Discussion. I personally find that the contents of these two sections are a little bit similar. Experimental Section should include all the details about experiments and characterizations, other than results, analysis and discussion. Results and Discussion should focus on the analysis and discussion of obtained experimental results, not the details about the experiments.
Authors Response: We checked the manuscript and think that, apart the previous section 3.1, that has now been taken out and it contents added to section 2.1, we do not find any other duplication of information between the contents of sections 2 (experimental) and 3 (results and discussion). Section 2.6 was added.
2.1. Synthesis
Cyclo-glycyl-L-alanine (1.29 g, 10 mmol), was dissolved in 5 ml of HI (57%, stabilized with H3PO3) and 10 ml of water. H3PO3 acted as a stabilizer for HI to avoid reduction to elemental iodine. After two weeks of slow evaporation at room temperature, transparent, hexagonal-shaped single crystals of Gly-L-Ala.HI.H2O (Poly2) were formed. The crystals were collected and rinsed with acetone, dried and kept in a dissector. An example of the grown crystals is shown in Figure 1. Cyclo-Glycyl-L-alanine (cyclo-Gly-L-Ala) was purchased from Bachem AG (Bubendorf, Switzerland). Hydriodic acid (HI) was purchased from Merck (Darmstadt, Germany) and used as received.
The synthesis of Gly-L-Ala.HI.H2O (Poly2) started with the cyclic form of the dipeptide (cyclo-Glycine-L-Alanine) and therefore the crystal growth conditions are different from those reported for Gly-L-Ala.HI.H2O (Poly1). We have also attempted the synthesis of linear glycyl-L-alanine with HI in an aqueous solution as described for Gly-L-Ala.HI.H2O (Poly2) using Cyclo Glycyl-L-alanine, but no crystals were ever formed. After complete evaporation of the solution, an oily residue was obtained. We conclude that Gly-L-Ala.HI.H2O (Poly1) is only obtained following the procedure reported in reference [1].
2.6. Piezoelectric Measurements
The piezoelectric properties of Gly-L-Ala.HI.H2O (Poly2) were analyzed by embedding the crystals within fibers fabricated by a conventional electrospinning technique described previously [32]. To produce the fibers, a clear and homogeneous 10 % polymer solution was prepared by dissolving 0.5 g of Poly (methyl methacrylate) (PMMA, Mw 996,000, Sigma-Aldrich, Schenlldorf, Germany) in 5 ml of chloroform. To this solution, 0.5 g of Gly-L-Ala.HI.H2O (Poly2) powder was added in a 1:1 weight ratio. The resulting mixture was stirred for several hours under ambient conditions before the electrospinning process. This precursor solution was loaded into a syringe, and its needle was connected to the anode of a high voltage power supply (Spellmann CZE2000). To produce in-plane fibers, the spinning voltage was set at 18 kV, with a distance of 12 cm between anode and collector. The flow rate of 0.10 mL/h was controlled by a syringe pump with an attached needle of 0.8 mm diameter. The fiber mat for piezoelectric measurements was collected on high purity aluminum foil, which served as electrodes.
The crystallinity and crystallographic orientation of Gly-L-Ala.HI.H2O (Poly2) inside the fibers were studied by XRD. The diffraction pattern was recorded between 5° and 50° using θ–2θ scans on a Philips PW-1710 X-Ray diffractometer with Cu-Kα radiation of wavelength 1.5406 Å. The morphology and fiber thickness were determined using a Nova NanoSEM scanning electron microscope operated at an accelerating voltage of 10 kV. Gly-L-Ala.HI.H2O (Poly2)@PMMA microfibers were deposited on a silica surface previously covered with a thin film (10 nm thickness) of Au-Pd (80–20 weight%) using a high-resolution sputter cover, 208 HR Cressington Company, coupled to an MTM-20 Cressington high-resolution thickness controller.
Piezoelectric output voltage and current were measured across a 100 MΩ load resistance connected to a low-pass filter, followed by a low-noise preamplifier (Research systems SR560, Stanford Research Systems, Stanford, California, USA), before being recorded with a digital storage oscilloscope (Agilent Technologies DS0-X-3012A, Waldbronn, Germany). The fiber array sample with a (30 × 40) mm2 area (200 µm thickness) was subjected to applied periodic mechanical forces imposed by a vibration generator (Frederiksen SF2185) with a frequency of 3 Hz determined by a signal generator (Hewlett Packard 33120A). The forces applied were calibrated using a force-sensing resistor (FSR402, Interlink Electronics Sensor Technology, Graefelfing, Germany). The sample was fixed on a stage, and the forces were applied uniformly and perpendicularly over the surface area.
- Reviewer: For the crystal structure as shown in Figure 3, 4, 5, 8, how do the author obtain these results? How to estimate the accuracy?
Authors Response: As written in section 2.2, the molecular and crystal structure diagrams, were drawn using the program Mercury [14] and the crystal data deposited in CCDC under deposit number 2247398, generally known as CIF file.
- Reviewer: From the results shown in Figure 7, the pyroelectric coefficient at room temperature is very small. Why? How to use this material as a pyroelectric or piezoelectric nanogenerator if the pyroelectric coefficient is so small?
Authors Response: For most of the pyroelectric materials, their pyroelectric coefficients for temperatures away from their Curie temperature when it exists, has always small values. Even for TGS (triglycine sulphate) crystals, a state-of-the art material for pyroelectric sensing, the pyroelectric coefficient at the ferroelectric-paraelectric transition temperature, is only one order of magnitude higher than that for the material under study..
- Reviewer: As this material is proved to be pyroelectric, then it is must a piezoelectric material. Could the authors provide characterizations about its piezoelectricity?
Authors Response A new section “3.6 Piezoelectric Response” about the characterization of the material as a piezoelectric active one has been added to the manuscript.
3.6. Piezoelectric Response
Any pyroelectric material is, by symmetry, also a piezoelectric and nonlinear optical material. Also, the tensor describing piezoelectricity and SHG properties of a crystalline material are the same and therefore for point group 2 the presented in SI6. Therefore, the suitability of Gly-L-Ala.HI.H2O (Poly2) crystals to be used as piezoelectric nangenerators for energy harvestings was investigated by embedding them into electrospun nanofibers as described before
An interconversion between a mechanical and an electrical stimulus arising due to an applied uniform stress, which generates electric polarization inside a dielectric material, is the origin of the piezoelectric effect. For crystalline solid to display the phenomenon, it must have a crystal structure without inversion symmetry. Gly-L-Ala.HI.H2O (Poly2) crystallizes in the polar point group 2, which is acentric. The tensor relationship between the polarization and stress , is given by the piezoelectric coefficient [43]. There is no preferential crystallographic orientation of the compound inside the electrospun fibers (see SI7). Therefore, a polarization develops under forces applied, repeatedly at regular times perpendicularly to the fiber array, and an effective piezoelectric modulus is measured along the same direction.
The applied stress (force per unit area) ranged between to . For Gly-L-Ala.HI.H2O (Poly2)@PMMA fiber mats, a applied periodical force originates, respectively, a and maximum instantaneous output piezoelectric voltage and current, as shown in Figure 16 a). Here, the two opposite peaks correspond to the press and release of the fiber mat. A plot of the output voltage as a function of several applied periodic forces shows an output voltage increasing linearly with the force magnitude as expected, Figure 16 b). PMMA polymer matrix is not piezoelectric with no contribution for the measure piezoelectric voltage. Considering a response time of , the magnitude of is obtained from the integration of the induced piezoelectric current over that period of time, , resulting in for Gly-L-Ala.HI.H2O (Poly2)@PMMA fiber mats. This induced charge, which is is related to the applied force by the equation,allows us to calculate an effective piezoelectric coefficient equal to . This value is of the same order of magnitude of that obtained for organic-inorganic ferroelectric perovskite trimethylchloromethyl ammonium trichloromanganese ((TMCM)MnCl3) where and barium titanate (BaTiO3) with [44,45]. It is also important to compare the present result, , with that obtained for lead-free organic–inorganic perovskite (N-methyl-N′-diazabicyclo[2.2.2]octonium)–ammonium triiodide (MDABCO-NH4I3) embedded into PMMA electrospun fibers (in a 1:5 ratio) which was reported to be MDABCO-NH4I3@PMMA [46].
Figure 16. (a) Piezoelectric current and output voltage as a function of time, with an applied force of 1N and a frequency of 3Hz. (b) Piezoelectric current as a function of different periodic forces applied, from Gly-L-Ala.HI.H2O (Poly2) incorporated into PMMA polymer microfibers.
It is remarkable that this new organic-inorganic Gly-L-Ala.HI.H2O (Poly2) crystalline compound exhibits, when embedded into electrospun fibers, a very high effective piezoelectric coefficient, similar in magnitude to an organic-inorganic perovskite containg also the iodide ion. In the present work, we demonstrate that Gly-L-Ala.HI.H2O (Poly2)@PMMA fibers may be incorporated in nanogenerators as active piezoelectric e materials.
Table 1. Piezoelectric nanogenerator parameters for some cyclic dipeptides and a semi-organic perovskite crystal.
|
Active compound
|
Force/area (N/m2) |
Vout (V) |
deff (pC/N) |
Power density (μWcm-2) |
Ref.
|
|
Gly-L-Ala.HI.H2O (Poly2)@PMMA(1:1) (Fiber mat) |
8 x102 |
28 |
280 |
0.65 |
This work
|
|
Cyclo(L-Trp-L-Trp)@PCL (1:5) (fiber mat) |
5 x103 |
9.6 |
30 |
0.13 |
[23]
|
|
Cyclo(GW) (crystal powder) |
7 x105 |
1.2 |
5.6*
|
0.002 |
[22]
|
|
Cyclo(FW) (crystal powder) |
6 x105 |
1.4 |
16** |
0.003 |
[21]
|
|
MDABCO-NH4I3@PMMA (1:5) (Fiber mat) |
11 x103 |
6.1 |
64 |
0.09 |
[46] |
* Calculated average piezoelectric coefficient from [22]; ** Calculated from data available in [21], assuming a nanogenerator time response of 0.5 s.
- Reviewer: Though the electrical conductivity increases a little when the temperature increases to over 350 K (but below 360 K), the conductivity value is still very small. Why? Would it have any influences on the performances of nanogenerators?
Authors Response: The studied material is an insulator and, due to this, it presents small values of the electric conductivity as experimentally observed. Regarding the influence on the performance of the nanogenerator, in the piezoelectric case, for example, the conductivity has a negative effect, since, as referred, a high conductivity increases the dielectric loss, tan(δ) and energy dissipation in the material. Additionally, in good conductors the free charges screening ability hinder the formation of the internal field driving the crystal deformation and appearance of the piezoelectric effect. Similar effects are present in pyroelectric materials. As such, good insulators are favoured for these nanogenerators.
- Reviewer: The authors present the optical second harmonic generation of this material. What is the relation of this property when using this material in nanogenerators? Since this manuscript is submitted to Special Issue: Piezoelectric Energy Harvesting and Sensing Technology: Materials, Mechanisms, and Applications, the authors are required to provide some related discussion about their proposed material’s possible application in piezoelectric energy harvesting and sensing.
Authors Response: A new section “Piezoelectric Response” about the characterization of the material as a piezoelectric energy harvester was added to the manuscript. Piezoelectricity and second harmonic generation are properties related to each other through the crystal point group, and described by the same tensor. However, their manifestation is entirely of different origin, as explained in the manuscript: second harmonic generation is a nonlinear optical property arising from the materials response to an intense optical field, while piezoelectricity is a response to an applied external mechanical force.
- The Conclusion Section is too long. The authors need to summarize the content.
Authors Response: The Conclusion section has been shortened as required.
- Conclusions
A polymorph (Gly-L-Ala.HI.H2O (Poly2)) of a previously reported Glycyl-L-alanine HI.H2O salt, was synthesized from the chiral cyclo-Glycyl-L-alanine dipeptide. The dipeptide is known to show molecular flexibility in different environments, which originated the polymorphism. The crystal structure of the Glycyl-L-alanine HI.H2O polymorph is determined at room temperature in space group P21, therefore it is a pyroelectric, piezoelectric and nonlinear optical material.
The pyroelectric coefficient reported in this work on a (010) orientated Gly-L-Ala.HI.H2O (Poly2) crystal plate, showed an increase with temperature with no significant abnormalities in the range 300-345 K reaching a maximum of at 345 K. Therefore, the pyroelectric coefficient of Poly2 is roughly three times higher than that reported to Gly-L-Ala.HI.H2O (Poly1) polymorph previously, which raised at 357 K. The different order of magnitude of the measured values for the two polymorphs results from the different atomic coordinates of the dipeptide, water molecules and the iodine ions within the unit cells of both compounds, which originates differences in some bond lengths magnitude and directionality of hydrogen bonds.
Thermal studies showed that Gly-L-Ala.HI.H2O (Poly2) starts degradation at 533 K, close to the melting temperature reported for cyclo-glycyl-L-alanine, which is 531 K. That temperature (533 K) is 30 K lower than that reported for linear glycyl-L-alanine dipeptide (563 K), suggesting that although the dipeptide when crystallized in Poly2 is not anymore in its cyclic form, it keeps a memory of its initial closed chain, therefore showing a thermal memory effect.
The DC conductivity behavior as a function of temperature in the regions below and above 350 K indicates in the low-temperature region characteristics of electrical conduction through the polaron transport behavior and at higher temperatures, which is characteristic of ionic conduction in the samples.
The efficiency of Gly-L-Ala.HI.H2O (Poly2) second harmonic generation was measured against a state-of-the-art nonlinear optical barium borate (BBO) crystal. A lower bound for the effective second-order nonlinear coefficient of Gly-L-Ala.HI.H2O (Poly2) crystals was estimated 0.14 pm/V.
Finally, an effective piezoelectric coefficient equal to was measured on an electrospun polymer fiber mat, Gly-L-Ala.HI.H2O (Poly2)@PMMA, demonstrating that the fibers are piezoelectric active systems with great potential to incorporated into energy harvesting devices.
- Overall, the organization and English expression of the manuscript need to be improved significantly. Please also carefully check out the whole manuscript to correct typos and grammar issues.
Authors Response: The English has been revised and corrected as required.

Round 2
Reviewer 2 Report
The authors have properly addressed the comments and therefore it is acceptable for publication in Materials.